# Analysis of Runoff-Sediment Cointegration and Uncertainty Relations at Different Temporal Scales in the Middle Reaches of the Yellow River, China

**Xiujie Wang, Dandan Li, Ximin Yuan \*, Xiling Qi and Pengfei Zhang**

State Key Laboratory of Hydraulic Engineering Simulation and Safety, Department of Hydraulic Engineering, School of Civil Engineering, Tianjin University, Tianjin 300350, China; wangxiujie@tju.edu.cn (X.W.); 2018205083@tju.edu.cn (D.L.); wangyanpeng@tju.edu.cn (X.Q.); gausswiiter@tju.edu.cn (P.Z.)
\* Correspondence: yxm@tju.edu.cn

**Abstract:** To understand the intricate runoff-sediment relationship in the middle Yellow River basin (MYRB), the Toudaoguai, Longmen, Tongguan and Huayuankou sites in the MYRB were selected to analyze the deterministic equilibrium and uncertainty relations of runoff-sediment based on 55-year hydrological data at multi-time scales. The Johansen test and wavelet neural network were used to verify the cointegration relationship among hydrological series. Runoff-sediment uncertain statistical relations and dynamics in the MYRB were also analyzed based on rating curves and hysteresis loops. The results showed that the logarithmic sequences of sediment load (SL), runoff and suspended sediment concentration (SSC) conformed to a linear cointegration relationship at the Toudaoguai station or in spring, winter or under small flow at other stations, but a nonlinear cointegration relationship was observed in other cases at other stations. Regarding runoff-sediment uncertain relationships, the rating curves, and hysteresis loops differed in stations (Toudaoguai and the other stations), as well as discharge (threshold: 1000 m$^3$/s), season (ice-flood and rainy season) and saturation of flow at flood and monthly scales. At the annual scale, phased and unsynchronized characteristics of runoff and sediment load were evident with a decreasing trend. This study on the runoff-sediment relationship can rationally provide a theoretical basis for the management and development of the Yellow River and other similar rivers with sufficient sediment, especially for areas with serious soil erosion.

**Keywords:** the middle Yellow River; cointegration relationship; uncertainty relationship; spatio-temporal scale; hysteresis loops; rating curves

## 1. Introduction

The Yellow River in China is famous for its heavy sediment load (SL), as well as its different sources of runoff and sediment and their intricate spatio-temporal changes, especially in the middle Yellow River basin (MYRB), which is endowed with unconsolidated and erodible loess, contributing 90% of the sediment load [1]. The behavior of sediment in the MYRB is an obvious nonlinear dynamic system, which is controlled by the complex interactions of time-varying sediment availability, rainfall intensity and distribution [2,3], as well as the morphology (slope, cross section shape) [4] and composition of the riverbed. Therefore, this implies an inharmonious and imbalanced relationship between runoff and sediment and complicated management issues of the Yellow River, such as the evolution of the channels [5,6], ecology and landscape [7,8] and the lifespan of infrastructure [9].

To date, the research on the Yellow River has mainly focused on particular areas with heavy sediment load for the regulation of Yellow River, including reaches of Ningxia–Inner Mongolia [10,11]

and the Hekouzhen–Longmen basin with abundant-coarse sediment [12,13], as well as main tributaries [6]. Due to the difference of grain size composition and geomorphologic agent, the regional characteristics of the Yellow River basin are obviously different. At present, there are few studies on runoff-sediment relationships across different characteristic regions. Due to the different sources of runoff and sediment, particle size composition and the effect of artificial regulation in the Yellow River basin, there is an obvious uncertain relationship between runoff and sediment, which has attracted the attention of numerous scholars [14–17]. In the study of the visualized runoff-sediment uncertain relationship during the process of sediment transport, a range of empirical models, such as sediment rating curves and hysteresis loops, are used to analyze and quantify suspended sediment concentration (SSC) and sediment track in rivers. Hysteresis patterns can provide useful insights into the sediment sources, sediment delivery and feedback mechanism [3,7,18]. Accurate estimates of sediment concentration by sediment rating curves are needed for erosion control structures, river morphological computations, management of ecosystems [2]. The regression parameters of the rating curve represent different physical meanings and explain the regional difference in sediment transport characteristics. For example, James et al. studied the correlations among SSC rating parameters, river basin morphology and climate [19]. The understanding of the uncertain relationship between runoff and sediment is of great significance to the construction of the water and sediment control system in the MYRB, the implementation of water resources management and configuration according to local conditions and the design of major water conservancy projects.

In previous research, the hydrological time series was generally assumed to be stable. However, the hydrological sequence generally includes long-term memory and is non-stationary [20]. Establishing a regression model based on non-stationary sequences for the runoff-sediment relationship is prone to the "pseudo-regression" problem [7]. Equilibrium relationships can be described as follows: the linear combination of non-stationary sequences is a stationary sequence [21]. It provides a scientific statistical method for non-stationary time series and interprets the long-term stable quantitative relationship among variables. The statistical relationship among variables with cointegration relationship no longer exists pseudo regression problem. Meanwhile, the error correction model (ECM), based on the cointegration relationship, can predict the hydrological variables effectively. Up to now, cointegration theory has been widely used in the field of econometrics, but it is rarely applied to watershed hydrology [22,23].

Focused on the above issues, this paper provides an in-depth study on the runoff-sediment stable equilibrium relation and uncertain relationship of Toudaoguai, Longmen, Tongguan and Huayuankou stations in the MYRB with different regional characteristics, followed by attempts to explore the hidden cointegration relationship, runoff-sediment dynamics, such as sediment origin and distribution, pollutant transport and the channel evolution and their statistical relation. This research rationally provides a theoretical basis for the management and development of the Yellow River and other similar rivers with sufficient sediment, especially for areas with serious soil erosion.

## 2. Study Area and Materials

The MYRB between the Toudaoguai and Huayuankou stations, located within 32°–42° N and 104°–113° E (Figure 1), significantly contributes to the total sediment of the Yellow River, due to the Loess Plateau. The area is characterized by loose soil, broken terrain, low vegetation coverage and frequent rainstorms [6,24]. Toudaoguai, Longmen, Tongguan and Huayuankou stations from upstream to downstream are distributed on the main stream of the MYRB, which locates in areas with different regional characteristics. The upstream reaches of the Toudaoguai are notable for their low sediment load and high water discharge. The reaches between the Toudaoguai and Longmen are mainly covered by coarse-grained and overburden sediment. From the Longmen to Sanmenxia, the river is characterized by high sediment load and fine sediment, and the reaches between the Sanmenxia and Huayuankou, including the Yiluo River and Qing River basins, have low sediment loads [25]. The length of this segment of the river is 1206.4 km (749.6 miles), which accounts for

22.1% of the total length of the river. The MYRB is 344,000 square kilometers (132,819 square miles), accounting for 45.7% of the total river basin [26]. Meanwhile, it accounts for 89% of the annual average sediment transport, and 90% of which comes from the rainy seasons. Tributaries with abundant sediment in the MYRB are plentiful, which carry an average of 56% of the river's annual sediment load. In addition, its annual mean temperature ranges from about 6 to 11 °C [26], and the spatial and temporal distribution of precipitation within the basin is uneven. Annual average precipitation ranges from 320 mm (12.6 inches) in the north to 836 mm (32.9 inches) in the south, and the potential evapotranspiration ranges from 810 (31.8 inches) to 1260 mm (49.6 inches). To meet the needs of flood control, power generation, navigation and irrigation, a series of reservoirs including Longyangxia (1986), Liujiaxia (1968), Qingtongxia (1960), Sanmenxia (1960) and Xiaolangdi (1997) reservoirs from upstream to downstream were put into operation, which greatly change the runoff-sediment conditions.

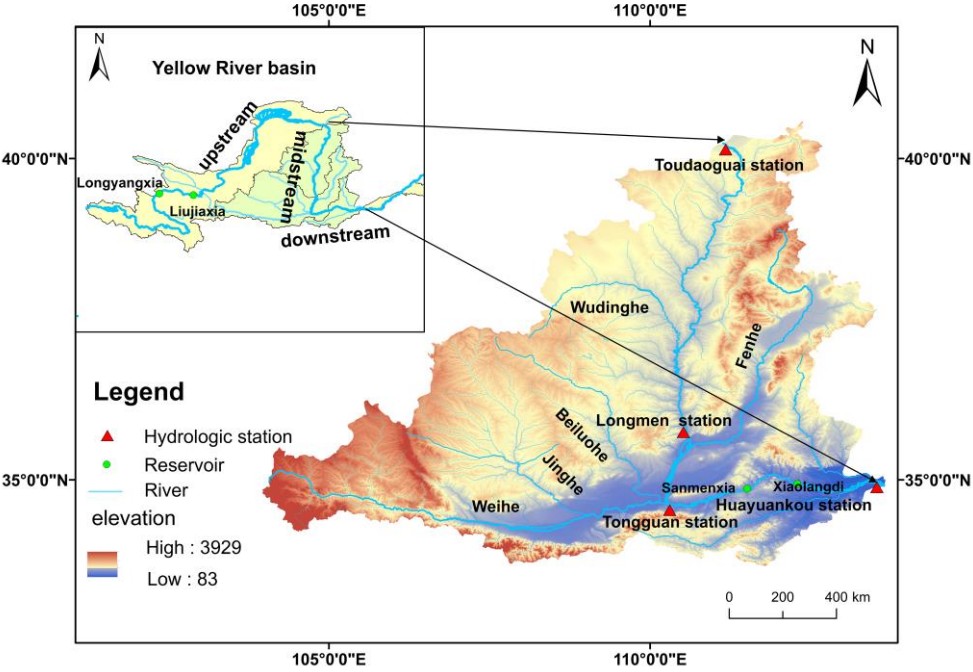

**Figure 1.** Schematic diagram of the middle reaches of the Yellow River.

The data used in this paper includes water discharge and SSC series of total 285 floods from 1960 to 2005 and corresponding daily series (including sediment transport rate) from 1960 to 2015 of the four stations. Based on the existing data, the homologous monthly and annual data of discharge, runoff, SSC and sediment load (suspended load and bed load) could be calculated. The data series were mostly collected from the Yellow River Water Conservancy Commission (YRCC) of the Ministry of Water Resources and the Yellow River Sediment Bulletin, which were all measured based on the Chinese national standard criterion.

## 3. Methods

### 3.1. Cointegration Theory

Non-stationary series are universal, which limits the use of the classical regression equation. Cointegration analysis is a statistical property that describes the long-term equilibrium relationship among time series and first proposed by Engle and Granger in 1978 to avoid the pseudo-regression problem among non-stationary sequences [21]. Its exact definition can be expressed as follows: if a time series becomes a stationary sequence after $d$ differences, then the original sequence is called a $d$-order single integer sequence, which is recorded as I($d$). If multiple time series $X_1, \cdots, X_n$ are all of the same order I($d$), and there is a vector $(\alpha_1, \cdots, \alpha_n)$ that makes the weighted combination $\alpha_1 X_1 + \cdots + \alpha_n X_n$

be a stationary sequence I(0), then this set of time series is cointegrated, which excludes the random trend caused by the unit root of the variable. The exploration of the cointegration relationship can reveal the long-term quantitative relationship among variables and avoid the pseudo-regression problem when it comes to statistical relationships among variables in a cointegration relationship. Meanwhile, the ECM based on cointegration relationship distinguishes between long-term equilibrium relations and short-term volatility relations, which can automatically correct the early errors and predict variables. Common cointegration test methods include the Engle-Granger (EG) two-step test (two variables) [21] and the Johansen test (more than two variables) [27]. The EG two-step test is a stationarity test based on regression residuals. The Johansen test was proposed by Johansen and Juselius to test the cointegration relationships of multi-variables with maximum likelihood estimation based on the VAR model. This study involves three variables, and there may be more than one cointegration relationship. Therefore, the Johansen test is used in this work to verify the cointegration relationship and determine the cointegration equation. Its specific principle is as follows [28]:

Presume that the vector $y_t$ is a $p$-order autoregressive model (VAR):

$$y_t = A_1 y_{t-1} + \cdots + A_p y_{t-p} + B x_t + \varepsilon_t \tag{1}$$

Then, convert this VAR into the following form of vector error correction model (VECM):

$$\Delta y_t = \Pi y_{t-1} + \sum_{i=1}^{p-1} \Gamma_i \Delta y_{t-i} + B x_t + \varepsilon_t \tag{2}$$

where $\Pi = \sum_{i=1}^{p} A_i - I$, $\Gamma_i = -\sum_{j=i+1}^{p} A_j$. If $0 < \text{rank}(\Pi) < n$, there are two matrixs $\alpha$ and $\beta$ whose order are all $n \times r$. makes $\Pi = \alpha\beta'$. and $\Pi = \alpha\beta'$ is I (0). When the rank of the matrix is r, then the matrix has r non-zero eigenroots, which is arranged by values as $\lambda_1, \lambda_2, \cdots, \lambda_r$. The significance test for the number of eigenroots can be performed by the trace statistic $\lambda_{trace}$ and the maximum eigenroot statistic $\lambda_{max}$.

$$\lambda_{trace} = -T \sum_{i=r+1}^{m} log(1 - \lambda_i) \tag{3}$$

$$\lambda_{max} = -Tlog(1 - \lambda_r + 1). \tag{4}$$

where $\lambda_i$ represents the estimate of eigenroots; $T$ represents the sample size.

The relationship can be called "linear cointegration" if the time series satisfy a long-term linear equilibrium relationship [29,30]. According to the cointegration theory, an ECM can be established to describe the long-term static performance and short-term dynamic characteristics among the sequences. A first-order autoregressive distribution lag model ADL (1, 1) with only two variables was considered. The expression of the error correction model is as follows:

$$\Delta y_t = \beta_0 + \beta_1 \Delta x_t + \alpha Ecm(t - 1) + \varepsilon_t \tag{5}$$

where $\Delta y_t$ represents the short-term fluctuations of the explained variable; $\Delta x_t$ represents the short-term fluctuations of the explanatory variables; $Ecm(t-1)$ is the error correction term at the previous moment, reflecting the degree to which the previous term of the dependent variable deviates from its long-term equilibrium state in short-term fluctuations; $\alpha$ is the correction coefficient, representing the adjustment speed; $\varepsilon_t$ represents residuals; $\beta_0$ and $\beta_1$ are the regression coefficients.

Hurst discovered that hydrological time series has long-term memory characteristics in 1951 [31], showing that there is a nonlinear relationship between hydrological time series. In addition, many hydrological time series have fractal dimensions, so the long-term equilibrium relationship is nonlinear. For the vector time series $X_t = (x_{1t}, x_{2t}, \cdots, x_{pt})^T$, the sequence of $\{X_t\}$ components is called nonlinear cointegration if (1) $x_{it}(i = 1, \cdots, p)$ is a long-term memory mean (LMM) sequence, and (2)

there is a nonlinear function $f$ such that $Z_t = f(x_{1t}, x_{2t}, \cdots, x_{pt})$ is a short-term memory mean (SMM) sequence. The function $f()$ is called a nonlinear cointegration function. The wavelet neural network (WNN), a typical nonlinear function, has been usually used to compute nonlinear cointegration function [32]. The WNN model can be expressed according to the following formula:

$$y_t = \sum_{n=1}^{N} \omega_n h((\sum_{m=1}^{M} v_{nm} x_{mt} - b_n)/a_n). \tag{6}$$

where $M$ is the number of nodes of the input layer; $N$ is the number of nodes of hidden layer; $x_{mt}$ is the input vector; the base function $h((x - b_n)/a_n)$ ($n = 1, 2, \ldots N$) is the wavelet function; $a_n$ is the expansion parameter and $b_n$ is the translation parameter; $v_{nm}$ is the connection weight from input layer to hidden layer; $\omega_n$ is the connection weight from hidden layer to the output. $y_t$ is the output value.

### 3.2. Methodology for Runoff-Sediment Relationship

A sediment rating curve, reflecting the runoff-sediment relationship, describes the relation between discharge (Q) and SSC for a certain location of the river through a mathematical model [33,34]. Sediment rating curves based on the existence of cointegration relationship can avoid spurious regression in the study of runoff-sediment relationship and truly reflect the actual relationship among hydrological variables. The power function is usually applicable for runoff-sediment relation when it is unsaturated flow. Once the flow is saturated, the sediment rating curves (SRC) between maximum SSC and Q emerges with linear form (Equation (7)) [2].

$$SSC = c(Q - Q_t) + d \tag{7}$$

where $c$ and $d$ are fitting coefficients, and $Q_t$ represents the threshold discharge between unsaturated and saturated flows.

The hysteresis loop with clockwise, anticlockwise or figure-eight shape is also introduced to interpret runoff-sediment relationship. The empirically-based approach which compares the simultaneity or interval between the SSC peak and discharge maximum is widely applied [5,9].

## 4. Results and Discussion

### 4.1. Runoff-Sediment Cointegration Equilibrium Relation

To reduce the absolute value of the data, eliminate the heteroscedasticity and facilitate calculation, logarithmic data were used for the cointegration calculation. The annual data of SL, runoff and SSC from 1960 to 2000, as an example, were first used for the study of the cointegration relationship among variables. Evidently, there was a good positive correlation between discharge and SSC at the Toudaoguai station at an annual scale, and a facial linear relationship existed among the logarithmic hydrological sequences of SL, runoff and SSC. This is in agreement with Fan's conclusion that there was a synchronous decrease of water and sediment at the Toudaoguai station, and SSC-Q fitted the power function when Q < 2000 m$^3$/s [5,35]. However, at the other stations, obvious changes of SSC did not correspond with the discharge due to a sharp increase of sediment in the MYRB [36,37], and no obvious statistical model existed for the scattered data.

#### 4.1.1. Linear Cointegration Relationship at Toudaoguai Station

In order to avoid problems such as pseudo-regression, it is necessary to verify the stationarity of the sequence before testing the cointegration relationship among the variables. The ADF (Augmented Dickey-Fuller) test was used to test the unit root of each time series at the Toudaoguai station. The test results are shown in Table 1.

**Table 1.** Unit root test results of sediment load (SL), runoff, and suspended sediment concentration (SSC) sequences.

| Sequences | Type | Null Hypothesis | ADF Value | %5 Threshold | *P* Value of ADF Statistic | *P* Value of Trend Item | Conclusion |
|---|---|---|---|---|---|---|---|
| SL | Trend, intercept | exist unit root | −5.49 | −3.52 | 0.0003 | 0.0004 | Trendy, stationary |
| runoff | Trend, intercept | exist unit root | −4.78 | −3.52 | 0.0020 | 0.0035 | Trendy, stationary |
| SSC | Trend, intercept | exist unit root | −5.49 | −3.52 | 0.0003 | 0.0002 | Trendy, stationary |
| D(SL) [1] | none | exist unit root | −8.39 | −1.95 | 0.0000 | — | stationary |
| D(runoff) | none | exist unit root | −8.23 | −1.95 | 0.0000 | — | stationary |
| D(SSC) | none | exist unit root | −8.00 | −1.95 | 0.0000 | — | stationary |

[1] D() represents the first order difference of the corresponding variable.

It can be seen from the unit root test results in Table 1, although the ADF statistics of the sequences SL, runoff and SSC were all less than the 5% threshold, the *P* values of the trend items were also all less than 0.05, which illustrated that the above three series at the Toudaoguai station were all stationary with a decreasing tendency. As Zhao et al. and Wang et al. explained that the reduced rainfall, reservoir regulation, increased water use for industry and agriculture and the effect of water and soil conservation could induce the downward trend [24,38]. After a first-order difference, it was determined that the three sequences at the "none case" were all stationary according to the principle of minimum AIC (Akaike Information Criterion) and SC (Schwarz Criterion) values. This implied that the time series were all subordinated to I(1). Thus, it was necessary to conduct the Johansen test on a non-stationary time series SL, runoff and SSC, to investigate whether there was a long-term stable equilibrium relationship.

Before conducting cointegration tests on non-stationary hydrological sequences, a vector autoregressive model needed to be performed objectively to determine the maximum lag period of the relevant variables. The values of each measurement criterion with different lag orders are given in Table 2, indicating that the optimal lag order was an order of 1.

**Table 2.** P-step lag values of AIC [1], SC [2] and HQ [3] statistics.

| Lag | AIC | SC | HQ |
|---|---|---|---|
| 1 | −6.11 | −5.72 | −5.97 |
| 2 | −6.00 | −5.22 | −5.72 |
| 3 | −5.94 | −4.78 | −5.53 |
| 4 | −6.05 | −4.49 | −5.49 |
| 5 | −5.98 | −4.04 | −5.29 |
| 6 | −6.10 | −3.77 | −5.27 |

[1] AIC represents the Akaike Information Criterion; [2] SC represents the Schwarz Criterion; [3] HQ represents the Hannan-quinn Criterion.

After determining the optimal lag order of the related variables, the Johansen cointegration test was used to verify the existence of long-term stable equilibrium relationships. The results of the cointegration test are shown in Table 3.

**Table 3.** The results of the Johansen cointegration test.

| Hypothesized No. of CE(s) (Cointegration Equation(s)) | Trace Test | | | Maximum Eigenvalue | | |
| --- | --- | --- | --- | --- | --- | --- |
| | Trace Statistic | 0.05 Critical Value | *p* Value | Max-Eigen Statistic | 0.05 Critical Value | *p* Value |
| None | 58.12 | 29.80 | 0.0000 | 30.84 | 21.13 | 0.0016 |
| At most 1 | 27.29 | 15.49 | 0.0006 | 16.80 | 14.26 | 0.0194 |
| At most 2 | 10.49 | 3.84 | 0.0012 | 10.49 | 3.84 | 0.0012 |

The *P* values obtained by the trace test and maximum eigenvalue test of three hypotheses of the number of cointegration relations were all less than 0.05. Thus, all three cases rejected the null hypothesis, which indicated that there were at least three cointegration equations among the variables. The following are the estimated cointegration relationships among the three variables.

$$\begin{cases} SL = 0.533 \, runoff + 1.429 \, SSC - 1.865 \\ SL = 1.916 \, SSC - 0.807 \\ SL = 2.125 \, runoff - 5.036 \\ runoff = 0.914 \, SSC + 1.985 \end{cases} \tag{8}$$

The Johansen cointegration test results indicated three sequences, and all pairs of variables had long-term linear cointegration relationships; i.e., at the Toudaoguai station, runoff and SSC had long-term stable quantitative influence on SL. Furthermore, there was also a stable association between runoff and SSC. This was similar to the research results that there was a good positive correlation between discharge and SSC at the Toudaoguai station. Therefore, the linear cointegration relationship among the variables of the Toudaoguai station is also applicable at other time scales. The determination of the quantitative relationship of the three variables is helpful to the calculation and prediction of hydrological series at the Toudaoguai station. Meanwhile, the existence of the cointegration relationship among the variables also indicated that there was no pseudo-regression problem concerning the study of the uncertainty relationships between them. In addition, the existence of the linear cointegration relationship paves the way for the establishment of the error correction model for the more accurate prediction of variables. Annual SL, runoff and SSC data from 1960 to 2000 were used to perform the Johansen cointegration test and calibrate the error correction model (ECM). Annual data from 2001 to 2005 were used to verify the parameters of ECM. The estimation formulas of the error correction models are shown in equation 5. Moosa et al. pointed out that the coefficient of the difference term indicates the degree and direction of short-term fluctuations on the short-term changes of variables, and the coefficient of the correction term indicates the strength of adjusting the short-term fluctuations from the equilibrium state variable until the equilibrium state is reached [39].

$$\begin{cases} ECM1: \Delta SL = 1.101 \, \Delta SSC + 0.917 \, \Delta runoff - 0.838 \, Ecm(-1) + 0.0004 \\ ECM2: \Delta SL = 1.813 \, \Delta SSC - 0.876 \, Ecm(-1) + 0.0022 \\ ECM3: \Delta SL = 2.116 \, \Delta runoff - 0.584 \, Ecm(-1) - 0.0015 \\ ECM4: \Delta runoff = 0.780 \, \Delta SSC - 0.595 \, Ecm(-1) + 0.0020 \end{cases} \tag{9}$$

During the calibration period (1960–2000), the Nash coefficients between the calculated values of ECM and measured values were 0.96, 0.93, 0.88 and 0.81, respectively. The corresponding Nash coefficients between the calculated values of the ordinary least squares (OLS) and measured values were 0.95, 0.91, 0.84 and 0.75, respectively. The calibration results indicated that the calculation accuracy of ECM was significantly improved compared with OLS. Christoffersen et al. provided an explanation as to why the addition of the error correction term leads to the enhancement of forecasting [40]. Likewise, Jacobson et al. also showed that the performance of the ECM was better if long-run theory restrictions were imposed on the cointegrating relation [41]. Therefore, the use of the ECM is recommended to

predict the hydrological variables, such as SL or runoff at the Toudaoguai station. More accurate prediction of runoff and sediment is of great significance to the flood control and disaster reduction, rational allocation of water resources, the solution of soil erosion and river deposition in the river basin. Meanwhile, the cointegration relationship among the three variables was the strongest with the highest Nash coefficient, and the relatively poor cointegration relationship between runoff and SSC was obvious. This was mainly dominated by the uncertain relationship between discharge and SSC, due to the incongruous relationship with reservoir regulation and the availability of sediment [35,42]. During the validation period (2001–2005), the comparative calculation results between the four ECMs and the OLS equations are shown in Tables 4 and 5, the prediction of the four ECMs was better than those of the OLS except for a few years. Meanwhile, the error of ECM based on three variables was smaller than that of the other models on average in the prediction of SL, which is consistent with the results of the calibration period. Ultimately, the prediction errors were less than 20% except for ECM 2 in 2002, ECM1 and ECM3 in 2004, and the minimum prediction error was 0.36%, which verified the good prediction ability of ECM.

### 4.1.2. Nonlinear Cointegration Relationship at the Other Stations

Prior to examining nonlinear cointegration, the R/S statistic method was used to determine the long-term memory characteristics of the annual average SL, runoff and SSC of the Longmen, Tongguan and Huayuankou stations from 1987 to 2005. Hurst exponents are given in Table 6.

It can be seen from Table 6 that the Hurst index values were more than 0.5 for all time series at three stations; thus, they were all LMM time series. The empirical formula of the fractal dimension D is D = 2 − H; the fractal dimensions of the original sequences are shown in Table 7.

Table 7 clearly shows that the fractal dimensions of the three hydrological sequences at each site were different; thus, which meant none of the sequences conformed to a linear cointegration relationship [20]. Therefore, the wavelet neural network (WNN), served as a fitting method, was used to verify the existence of nonlinear cointegration among variables [22,23]. With the WNN method, WNNs with a single hidden layer were established, and the structure of WNN was 3–8–1, 2–8–1, 2–8–1, 2–8–1 for the input variables SL–runoff–SSC, SL–runoff, SL–SSC, and runoff–SSC, respectively. A short-term memory mean (SMM) zero time series served as the guidance output. The mallet wavelet was adopted for the wavelet base function. The training results showed that the function $f(\cdot)$ existed in all of the the above training combinations, which indicated that $y_t = f(x_{1t}, x_{2t}, \cdots, x_{nt})$ was a SMM time series. In other words, all of the above combinations showed nonlinear cointegration. The relevant parameters of the nonlinear cointegration function of different training combinations are shown in Tables 8–11. Since the statistical relationships between hydrological series at the Longmen, Tongguan and Huayuankou stations were not significant and their nonlinear relationships were strong, the ECM based on nonlinear cointegration relationship was not studied in depth in this paper.

**Table 4.** Predicted results of sediment transport at Toudaoguai station.

| Year | Measured Value | OLS(Three Variables) [1] | | ECM1(Three Variables) [2] | | OLS(SL, SSC) [3] | | ECM2(SL, SSC) [4] | | OLS(SL, Runoff) | | ECM3(SL, Runoff) | |
|---|---|---|---|---|---|---|---|---|---|---|---|---|---|
| | | Calculated Value | Re [5] | Calculated Value | Re | Calculated Value | Re | Calculated Value | Re | Calculated Value | Re | Calculated Value | Re |
| 2001 | 0.20 | 0.18 | 9.12% | **0.18** | **9.92%** | 0.17 | 14.66% | **0.18** | **10.84%** | 0.23 | 13.57% | **0.20** | **0.36%** |
| 2002 | 0.27 | 0.33 | 24.29% | **0.31** | **14.40%** | 0.37 | 35.65% | **0.36** | **33.45%** | 0.25 | 6.72% | **0.25** | **7.46%** |
| 2003 | 0.28 | 0.29 | 3.98% | **0.28** | **0.61%** | 0.32 | 13.02% | **0.31** | **10.24%** | 0.22 | 20.43% | **0.23** | **18.64%** |
| 2004 | 0.24 | 0.28 | 17.26% | **0.30** | **23.62%** | 0.28 | 17.33% | **0.28** | **16.90%** | 0.27 | 14.64% | **0.30** | **24.69%** |
| 2005 | 0.40 | 0.45 | 12.57% | **0.43** | **6.74%** | 0.48 | 18.21% | **0.46** | **13.19%** | 0.37 | 8.91% | **0.37** | **9.65%** |

[1] OLS (three variables) represents the ordinary least squares (OLS) regression model among three variables; [2] ECM1 (three variables) represents the ECM1 among three variables; [3] OLS (SL, SSC) represents the OLS model between SL and SSC; [4] ECM2 (SL, SSC) represents the ECM2 between SL and SSC; [5] Re represents the relative error between variables. The other parameters have similar meanings.

**Table 5.** Predicted results of runoff at Toudaoguai station.

| Year | Measured Value | OLS(Runoff, SSC) | | ECM4(Runoff, SSC) | |
|---|---|---|---|---|---|
| | | Calculated Value | Re | Calculated Value | Re |
| 2001 | 113.28 | 100.65 | 11.15% | **110.70** | **2.28%** |
| 2002 | 122.75 | 144.66 | 17.85% | **144.58** | **17.78%** |
| 2003 | 115.57 | 135.14 | 16.93% | **128.31** | **11.02%** |
| 2004 | 127.61 | 127.75 | 0.11% | **121.47** | **4.81%** |
| 2005 | 150.21 | 164.64 | 9.61% | **159.32** | **6.07%** |

**Table 6.** Hurst exponents of hydrological series SL, runoff, SSC.

| Sequence | SL | Runoff | SSC |
|---|---|---|---|
| Longmen | 0.66 | 0.71 | 0.63 |
| Tongguan | 0.68 | 0.92 | 0.55 |
| Huanyuankou | 0.79 | 0.91 | 0.89 |

**Table 7.** Fractal dimension of hydrological series SL, runoff, SSC.

| Sequence | SL | Runoff | SSC |
|---|---|---|---|
| Longmen | 1.34 | 1.29 | 1.37 |
| Tongguan | 1.32 | 1.08 | 1.45 |
| Huanyuankou | 1.21 | 1.09 | 1.11 |

**Table 8.** Related parameters of SL−runoff−SSC cointegration equation.

| Longmen | | | | | | Tongguan | | | | | | Huayuankou | | | | | |
|---|---|---|---|---|---|---|---|---|---|---|---|---|---|---|---|---|---|
| $a_n$[1] | $b_n$[2] | $\omega_n$[3] | $v_{nm}$[4] | | | $a_n$ | $b_n$ | $\omega_n$ | $v_{nm}$ | | | $a_n$ | $b_n$ | $\omega_n$ | $v_{nm}$ | | |
| 0.864 | −0.706 | 1.082 | −0.375 | 2.333 | −1.499 | 1.118 | 0.290 | 0.815 | 0.260 | 1.443 | −0.336 | −0.649 | −1.037 | 0.789 | 0.621 | 0.817 | 1.451 |
| 1.968 | −0.156 | −0.829 | −2.087 | 1.244 | −1.579 | 0.780 | 0.401 | −0.921 | −0.368 | −1.209 | −0.542 | 0.328 | −1.573 | 0.674 | 0.584 | −0.230 | −0.372 |
| 0.144 | −1.121 | −1.194 | 0.974 | 0.716 | −0.403 | 0.733 | −0.447 | −1.149 | −0.343 | −2.014 | 0.250 | 1.483 | −0.354 | 0.206 | −1.759 | 0.245 | 0.943 |
| −0.626 | −0.530 | 1.138 | 2.009 | −1.166 | −2.042 | 0.015 | −2.176 | 0.292 | −2.253 | −0.493 | −0.262 | −0.942 | −0.335 | −0.504 | −0.224 | −0.733 | 0.635 |
| 1.849 | −0.086 | −0.373 | 0.544 | 0.074 | 0.642 | −1.064 | −1.222 | −1.240 | −0.893 | 0.704 | −1.816 | −0.870 | −0.542 | −0.531 | −0.459 | −2.289 | −1.505 |
| 2.918 | −0.011 | 0.231 | −0.782 | −1.112 | −0.231 | 0.460 | 0.280 | 0.971 | 1.397 | −0.803 | 0.083 | −0.366 | −0.151 | −2.844 | −2.180 | 0.537 | 0.177 |
| 1.549 | −0.269 | 0.520 | 0.805 | −1.026 | 0.815 | −1.294 | 2.442 | −1.363 | −0.628 | −0.810 | −2.731 | 0.417 | −1.048 | −1.315 | −1.127 | −1.479 | 0.230 |
| −0.974 | 1.024 | 1.101 | −1.680 | −1.136 | −1.361 | 1.557 | 0.761 | 0.564 | 0.357 | −1.619 | 0.238 | −0.659 | 0.559 | 1.042 | −0.525 | 2.584 | 0.416 |

$h((x - b_n)/a_n)$ (n = 1, 2, … N) is the wavelet function; [1] $a_n$ is the expansion parameter; [2] $b_n$ is the translation parameter; [3] $\omega_n$ is the connection weight from hidden layer to the output; [4] $v_{nm}$ is the connection weight from input layer to hidden layer.

**Table 9.** Related parameters of SL−runoff cointegration equation.

| Longmen | | | | | Tongguan | | | | | Huayuankou | | | | |
|---|---|---|---|---|---|---|---|---|---|---|---|---|---|---|
| $a_n$ | $b_n$ | $\omega_n$ | $v_{nm}$ | | $a_n$ | $b_n$ | $\omega_n$ | $v_{nm}$ | | $a_n$ | $b_n$ | $\omega_n$ | $v_{nm}$ | |
| 1.105 | 0.645 | 0.427 | 1.165 | 0.834 | 0.515 | 0.149 | 0.382 | −0.903 | −1.331 | 1.037 | 0.024 | −0.885 | 0.437 | −1.881 |
| −0.520 | 0.441 | −0.347 | 1.136 | 0.117 | 0.178 | −0.376 | −0.832 | −0.189 | 0.174 | 1.134 | −0.573 | −0.715 | 0.467 | −0.557 |
| −0.199 | 0.566 | −1.803 | −0.207 | −0.273 | 0.095 | −1.193 | 1.198 | −0.325 | 0.854 | −1.243 | 0.752 | −1.895 | −1.593 | −1.623 |
| −1.799 | 1.835 | −1.121 | 0.756 | −2.132 | 0.415 | 0.675 | 0.891 | 2.124 | 0.351 | −0.801 | −0.670 | −1.117 | 1.222 | −1.236 |
| −0.860 | 0.521 | 0.784 | 1.630 | 0.573 | 1.035 | −0.287 | −0.074 | −0.606 | −0.776 | 0.159 | −0.298 | 0.618 | 0.197 | −0.217 |
| 0.717 | 0.388 | 0.742 | −0.847 | −0.377 | 5.803 | 0.004 | 0.267 | −0.325 | −1.547 | −0.380 | −0.750 | −1.139 | −1.693 | −0.373 |
| −0.065 | −0.095 | −1.098 | −0.540 | −1.677 | −0.454 | −0.715 | −0.848 | −1.614 | 2.228 | 1.159 | 0.108 | 1.883 | 1.297 | −0.327 |
| 0.839 | −1.327 | 0.556 | −1.702 | −1.024 | 0.953 | −1.569 | −0.533 | −0.644 | 2.832 | 1.442 | 1.056 | 0.529 | −1.654 | 0.080 |

The parameters have the same meaning as Table 8.

**Table 10.** Related parameters of SL−SSC cointegration equation.

| Longmen | | | | | Tongguan | | | | | Huayuankou | | | | |
|---|---|---|---|---|---|---|---|---|---|---|---|---|---|---|
| $a_n$ | $b_n$ | $\omega_n$ | $v_{nm}$ | | $a_n$ | $b_n$ | $\omega_n$ | $v_{nm}$ | | $a_n$ | $b_n$ | $\omega_n$ | $v_{nm}$ | |
| −0.125 | −0.098 | −1.171 | −0.048 | 1.304 | −0.949 | 0.326 | −0.875 | 3.251 | 0.736 | 0.877 | 0.942 | 2.041 | −0.922 | 0.627 |
| 0.362 | −0.250 | 0.513 | 1.592 | 2.143 | −0.462 | −0.445 | −1.033 | −0.258 | 1.316 | 1.095 | −0.836 | 0.860 | −0.940 | −0.872 |
| −0.270 | 0.773 | −1.944 | 1.065 | −0.892 | 3.345 | −0.269 | 1.797 | −0.590 | −4.218 | 0.258 | −0.438 | 0.795 | 0.403 | −0.477 |
| −0.378 | −0.439 | 0.942 | 0.218 | −1.966 | −0.697 | 0.863 | 1.900 | −0.271 | −0.336 | 0.543 | 2.995 | −2.532 | −2.595 | −0.096 |
| 0.028 | −0.773 | 0.395 | −1.320 | −0.126 | −0.522 | −0.733 | −1.514 | 1.362 | −0.208 | −0.686 | −0.608 | 0.792 | 0.255 | −0.518 |
| 1.396 | 0.411 | −0.136 | 0.829 | 1.435 | 0.819 | 1.251 | 2.607 | 0.208 | 0.701 | −1.107 | 1.934 | 0.633 | −0.831 | −0.369 |
| 0.001 | 0.002 | 1.051 | 0.547 | −0.198 | 0.354 | 0.851 | 1.702 | 0.758 | −0.443 | −0.356 | 1.048 | −0.484 | 1.163 | 1.099 |
| −0.783 | 1.215 | 0.053 | −0.971 | −0.703 | 0.896 | 0.270 | 0.224 | −0.974 | 0.352 | 0.198 | −0.546 | 0.940 | −0.958 | 0.707 |

The parameters have the same meaning as Table 8.

**Table 11.** Related parameters of runoff−SSC cointegration equation.

| Longmen | | | | | Tongguan | | | | | Huayuankou | | | | |
|---|---|---|---|---|---|---|---|---|---|---|---|---|---|---|
| $a_n$ | $b_n$ | $\omega_n$ | $v_{nm}$ | | $a_n$ | $b_n$ | $\omega_n$ | $v_{nm}$ | | $a_n$ | $b_n$ | $\omega_n$ | $v_{nm}$ | |
| 0.543 | −0.946 | −1.298 | 0.687 | 4.025 | 1.027 | 0.067 | 0.492 | −1.086 | 0.122 | −1.789 | 0.697 | −0.055 | 0.511 | −0.642 |
| 1.366 | −1.012 | 1.036 | 1.613 | 3.357 | −1.074 | 0.308 | −0.607 | −0.278 | 0.676 | −0.146 | 0.847 | −0.950 | −0.704 | −1.174 |
| 0.767 | −2.963 | 1.494 | −2.682 | −1.007 | 1.759 | 0.614 | 0.267 | 2.839 | 0.425 | 0.261 | 0.518 | 1.122 | 1.271 | −0.696 |
| −0.395 | 1.472 | 1.331 | 0.753 | 3.346 | −0.395 | −1.461 | −0.532 | −1.080 | 1.297 | 0.528 | 0.087 | −1.204 | 0.120 | −1.374 |
| 0.174 | 0.320 | 1.211 | 0.578 | 1.204 | −1.163 | −0.379 | 1.314 | 0.488 | −3.237 | 0.799 | 0.832 | −0.547 | −2.036 | 0.062 |
| −0.716 | −0.818 | −2.101 | −1.226 | 0.619 | −0.586 | −1.913 | −0.721 | −1.118 | −0.864 | 0.235 | −0.885 | −1.124 | −1.041 | 0.485 |
| 0.904 | 1.318 | 0.550 | 0.514 | 1.208 | 1.010 | −0.428 | −0.285 | 1.264 | 0.710 | −0.958 | 0.113 | 0.250 | −2.124 | −0.289 |
| −1.062 | −1.794 | 0.918 | 0.145 | −0.501 | 1.009 | 1.021 | −0.848 | 0.308 | −0.010 | −0.590 | 1.445 | 0.792 | 0.802 | 0.124 |

The parameters have the same meaning as Table 8.

### 4.2. Runoff-Sediment Uncertainty Relation

4.2.1. Runoff-Sediment Relation at the Within-Flood Event Scale

As shown in Figure 2a, the relation between discharge and SSC at the Toudaoguai station could be described by a cubic polynomial function at the within-event scale. This was explained from a macro perspective by Fan et al. [39] that the decreasing velocity (caused by overbank discharge) promotes the formation of sandbars, which narrows the channel width and accelerates the further scouring of the channel bed. Dissimilarly, at the other three stations, wind-dominated geomorphologic agents (geomorphologic agents here refer to external agents, which are mainly generated by solar energy, gravitational energy and biological activitiesgravity, including hydraulic force, wind force, ice force, wave force etc.) in the spring and winter transport coarse sediment to slopes, gullies and river courses [43]. However, coarse particles cannot enter the slurry, because of the lack of sufficient fine sediment and low energy flow [44], resulting in a lower SSC with respect to $Q < 1000$ m$^3$/s and emerging rating curves in the form of a power function. Simultaneously, the surface material composition of the MYRB has significant zonal characteristics: the composition is in the order of eolian sediment, sand loess covered by discontinuous aeolian sand, loess and clay loess from northwest to southeast [43,45], where the granularity composition gradually becomes finer. The high slurry viscosity of low energy flow for dominant fine sediment may form a laminar flow [46], which would lead to the synchronization of runoff and sediment. Therefore, the situation is similar to that of the Toudaoguai station at this time, and SL, runoff and SSC satisfy the linear cointegration relationship at the flood scale, and the same is true for other scales. When $Q > 1000$ m$^3$/s, energy flow with abundant fine sediment will continue to carry the coarse sediment carried by the wind in winter and spring downward with a high amplitude of concentration, which promotes the formation of hyperconcentrated flow. The linear cointegration relation no longer applied. The SRCs between the maximum SSC values for a given discharge displayed a linear form once the flow was saturated and stable, implying a global maximum with an optimal particle size composition ratio [47] (Figure 2b–d). The utilization of optimal particle size ratio and discharge threshold (threshold: $1000$ m$^3$/s) is beneficial to the dredging of reservoirs. It was deduced that the sediment composition of the Longmen station was coarser than that of the Tongguan station, because hydraulic erosion plays a dominant role in the southeast, where the surface composition was dominated by fine particles from loess [48]. Therefore, the maximum SSC was larger than that of the Tongguan station, due to the appropriate particle size composition. In addition, the SSC values in the third stage were all larger than those in the previous periods at the three stations due to the large-scale utilization of clean water resources [49,50] and enhanced scouring capacity for the regulation in runoff of Longyangxia (relatively weak regulation of sediment because of location) [51]. Meanwhile, the optimized particle size ratio for the enhanced coarse sediment-carrying capacity and abundant fine sediment produced by "trapping coarse and discharging fines" of reservoirs and water-and-soil conservation measures also facilitated the increase in the SSC.

Sediment rating curves can not explain the internal runoff-sediment relationship from a microscopic point of view; thus, the hysteresis loop was introduced to determine the hysteresis behavior according to an empirically-based approach, which can provide significant information on suspended sediment sources. Clockwise loops, which occur in late autumn and winter frequently [7,8], can be attributed to the available and deposited sediment source nearby, indicating sufficient available sediment and a quick increase of SSC at the beginning [52]. Meanwhile, overbank floods may be another reason for the reduction of SSC with higher discharge. Sediment from upstream, tributaries and hillslopes (gravitational erosion with gradual penetration) is a major reason for anti-clockwise hysteresis, which may suggest the delay of sediment entrainment within a runoff event [7]; thus, slumps and slides usually occur at the falling stage of a flood, which usually occurs in early spring, summer and early autumn [7,8]. Simultaneously, the intensive sediment-carrying capability of hyperconcentrated flow results in sediment with minimal deposition, even with a lower energy flow in the falling stage of a flood, which also gives rise to an anticlockwise hysteresis loop [53]. Figure-eight shapes

indicate more complex situations that mainly appear in early spring and summer [52]. To study the proportions of the three hysteresis types and the proportional changes of the same types at the four stations, 47, 62, 75 and 101 typical floods (an obvious flood process in which only one SSC peak occurs during a flow fluctuation) that occurred at the four stations were analyzed. It was observed that the clockwise, counter-clockwise, and figure-eight types accounted for 23.4%, 19.15% and 57.45% of floods at the Toudaoguai station, respectively. Correspondingly, the proportions of the three types at the Longmen, Tongguan and Huayuankou stations were 4.84%, 70.97% and 24.19%; 22.67%, 37.33% and 40%; and 19.80%, 45.54% and 34.65%, respectively. It was evident that counter-clockwise loops occurred at higher rates at the Longmen, Tongguan and Huayuankou stations, which was closely related to the massive sediment supply from abundant tributaries in the MYRB [24], the formation of the hyperconcentrated flow mentioned above and the sediment-delayed effect of soil and water conservation. The decrease of the proportion of anti-clockwise loops along the way was also related to the decrease of the formation frequency of hyperconcentrated flow with a predominance of fine particles and the decreased effect of soil and water conservation, which was verified by Xu et al. that the high frequency of the hyperconcentrated flow on the Loess Plateau occurred from northeast to southwest in space [54] and the sediment reduction of silt dam in the MYRB reached its peak in the 1970s, and has been declining ever since [55].

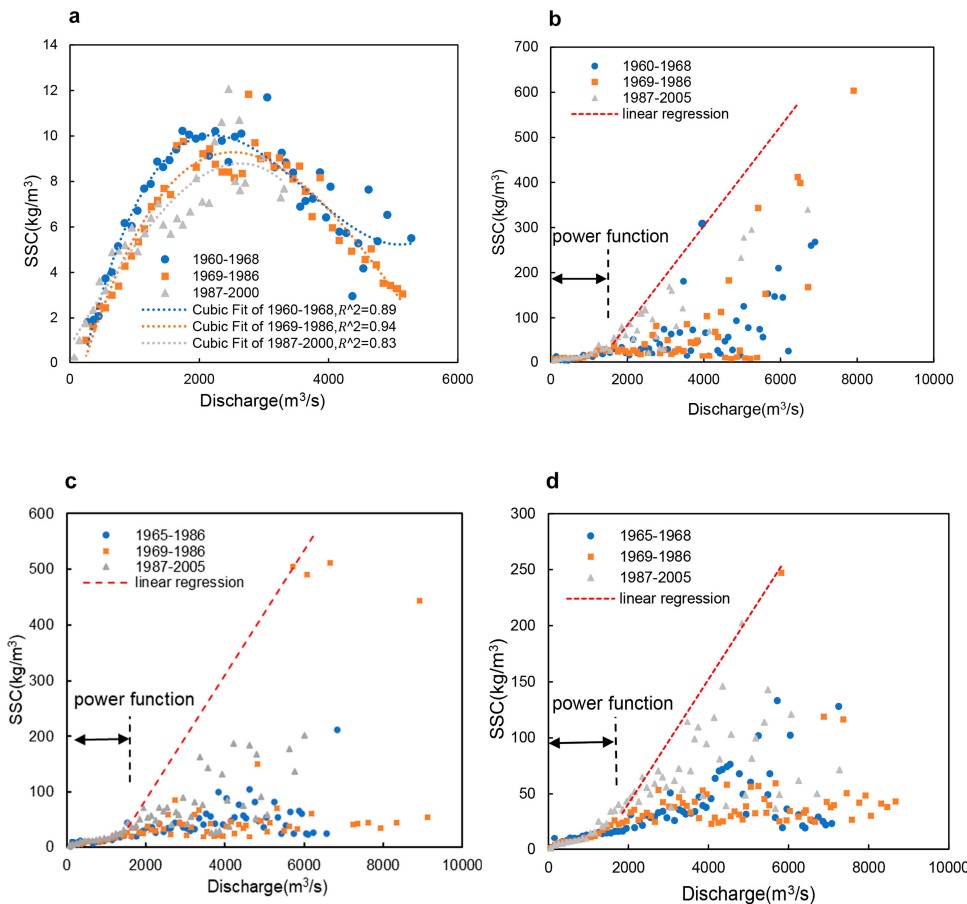

**Figure 2.** Rating curves for SSC and discharge at the (**a**) Toudaoguai, (**b**) Longmen, (**c**) Tongguan and (**d**) Huayuankou station during from 1960 to 2005.

### 4.2.2. Runoff-Sediment Relation at the Monthly-Seasonal Scale

Apparently, sediment rating curves varied seasonally at the four stations for the entire period. In the ice-flood season, the runoff, which was of a small amount, mainly originated from melting

ice, and the sediment condition was similar to that when $Q < 1000$ m³/s. Therefore, changes in SSC and Q were basically simultaneous, and the sediment rating curves derived from linear regression fitted the data well (Figure 3a,c,e,g). Specifically, the rating curve at the Toudaoguai station could be described by a cubic polynomial function with a high correlation coefficient (0.89) in rainy seasons (Figure 3b). Nevertheless, the correlations between SSC and Q were very low from the Longmen to Huayuankou stations due to the complex supply of sediment and the formation of hyperconcentrated flow with different combinations of coarse and fine sediment [43], shaping multiple SSC values for a given discharge (Figure 3d,f,h). Furthermore, the fluctuation frequency slowed down in the order of the Longmen, Tongguan and Huayuankou stations, which was determined by soil and water conservation measures, the operation of Xiaolangdi reservoir [56] and the relatively stable sediment composition [43,44] (dominated by fine particles) along the way.

At the monthly scale, the research period of the Toudaoguai station was divided into three stages with the same sudden change point of discharge and SSC caused by the operation of the Liujiaxia reservoir (1968) and the Longyangxia reservoir (1986) [5,35]. It was observed that hysteresis at the Toudaoguai station exhibited figure-eight patterns (Figure 4). Nevertheless, hysteresis loops with clockwise trends were found for the other stations (Figure 5). Analyses of the results showed that the sediment from the MYRB was much larger than that from upstream; in other words, the nearby suspended and deposited sediments were the main sediment source for the MYRB, which explained why the relations between Q and SSC showed hysteresis with clockwise trends at the other stations. Furthermore, the shapes of the C–Q hysteresis loops at the four stations evolved from linear loops at the Toudaoguai station to flattened loops with a sharp decrease of SSC and high discharge in September and October at the Longmen, Tongguan and Huayuankou stations, because the formation frequency of hyperconcentrated flow was greatly reduced due to the depletion of coarse sediment at the end of the rainy reason. Simultaneously, there were similarities; for example, the total suspended sediment load, affected by seasonal weather conditions, increased from May to August and fell dramatically from September to January [7].

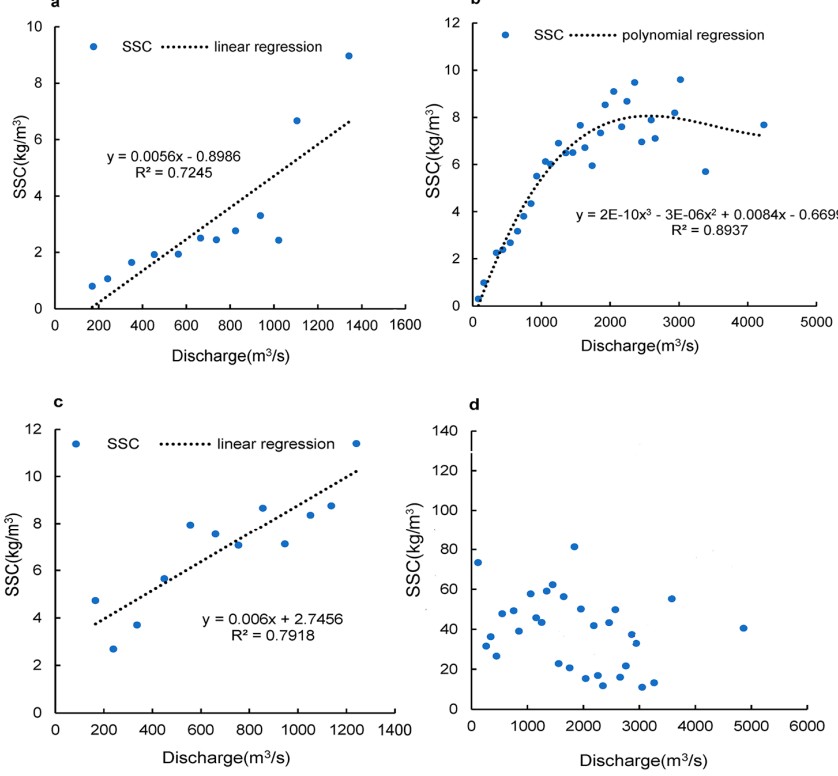

**Figure 3.** *Cont.*

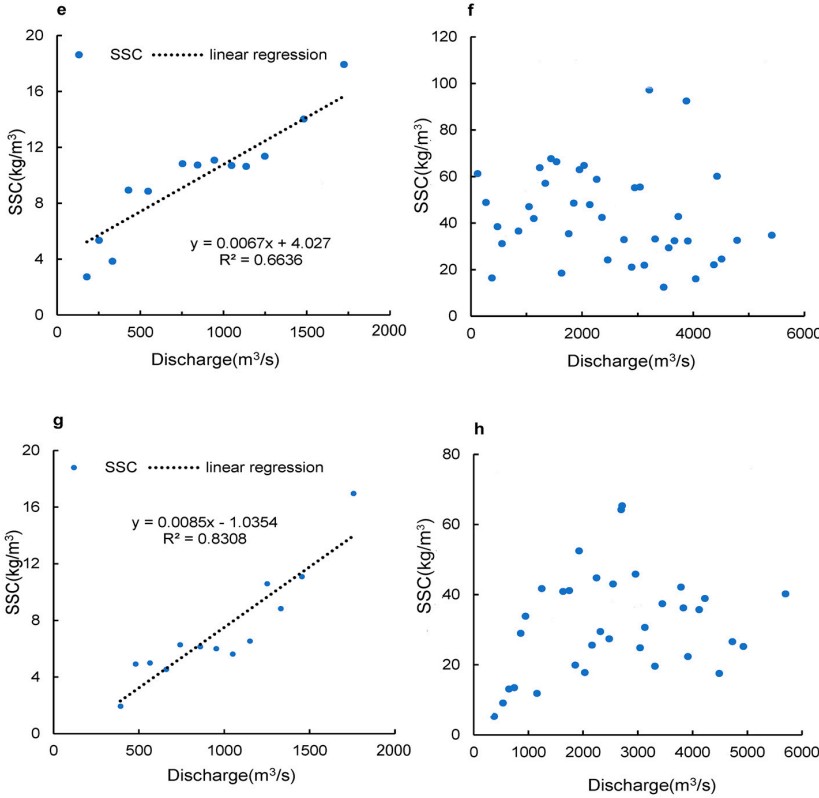

**Figure 3.** Seasonal rating curves for discharge and SSC in ice-flood seasons at (**a**) Toudaoguai, (**c**) Longmen, (**e**) Tongguan, (**g**) Huayuankou stations and in rainy seasons at (**b**) Toudaoguai, (**d**) Longmen, (**f**) Tongguan, (**h**) Huayuankou stations from 1960 to 2015.

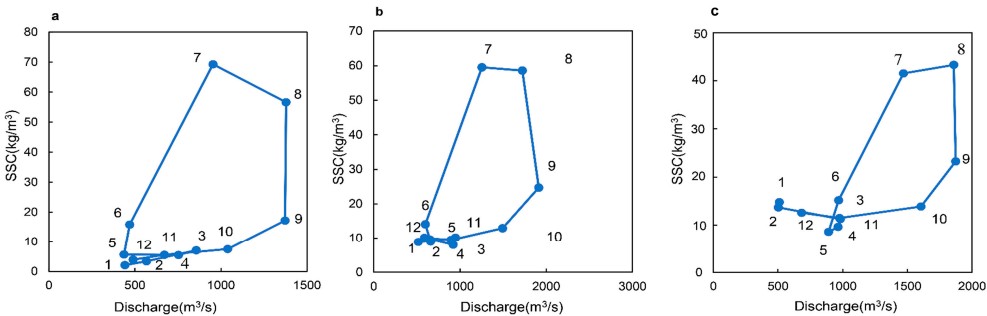

**Figure 4.** Monthly average hysteresis loop of SSC and discharge during the period of (**a**) 1960−1968, (**b**) 1969−1986, (**c**) 1987−2015 at Toudaoguai station.

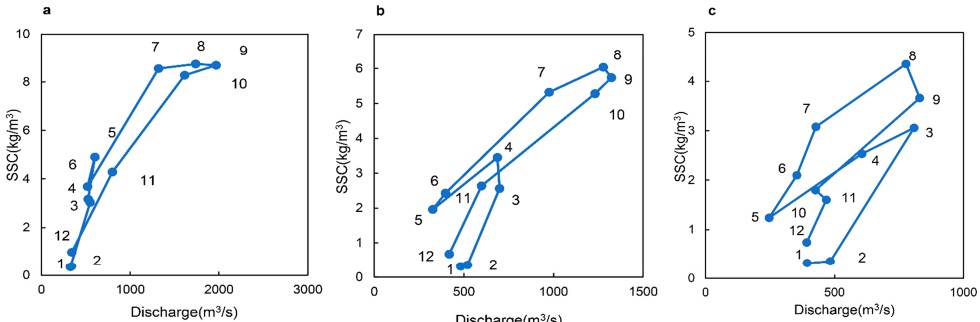

**Figure 5.** Monthly average hysteresis loop of SSC and discharge during the period of 1960−2015 at (**a**) Longmen, (**b**) Tonguan, (**c**) Huayuankou stations.

### 4.2.3. Runoff-Sediment Relation at the Annual Scale

It can be seen in Figure 6 that the variations of annual mean discharge, SSC and fluctuation amplitude at the four stations all displayed marked decreasing trends, and the current runoff and sediment were the driest stages in the runoff-sediment cycle [57]. In particular, the changes of SSC were basically synchronous with discharge at the Toudaoguai station. Nevertheless, this was not the case at other stations because the abundant sediment supply with a fragile Loess Plateau triggered the weak correlation between SSC and Q [35,37]. Spatially, the runoff mainly comes from the upper reaches, leading to the same variation discipline of the runoff at the four stations under the operation of the Liujiaxia in 1986 and the joint control of the Liujiaxia and Longyang in 1968, while the variation rules of sediment load, mainly from the middle reaches, were different. Natural causes—such as precipitation, temperature, vegetation and human activities, such as soil and water conservation projects, the operation of dams and reservoirs, water consumption and frequent tillage—could principally account for the changes in runoff and sediment reported from many investigations [24,58].

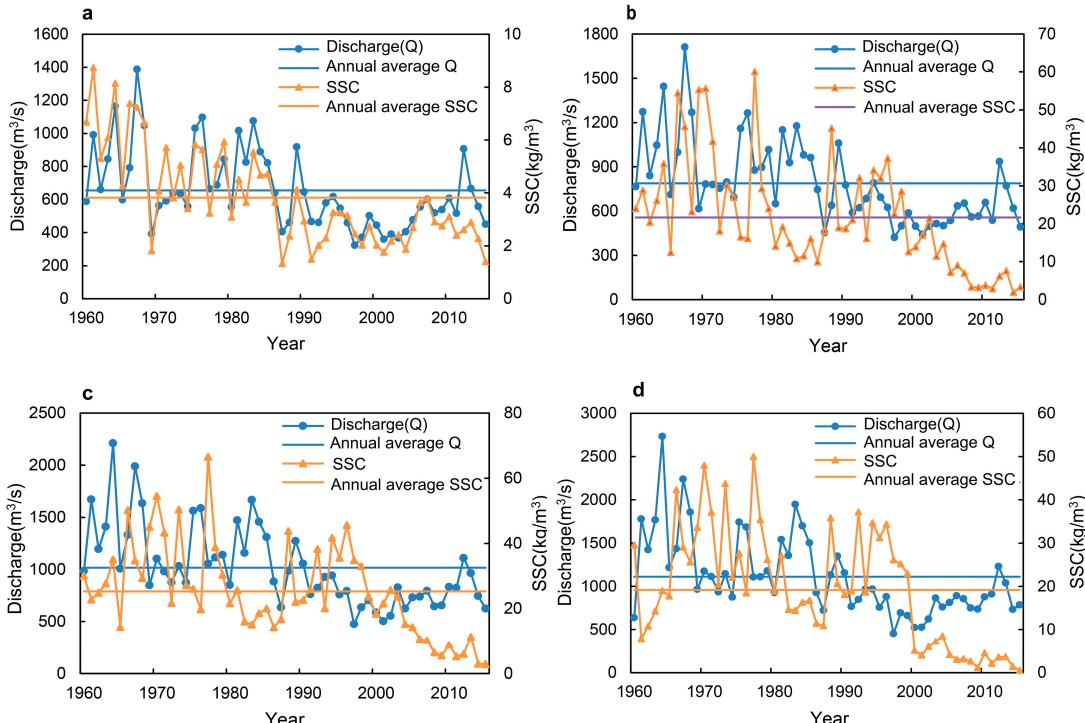

**Figure 6.** Changes in annual mean water discharge and suspended sediment concentration at Toudaoguai station.

Extreme precipitation or rainfall erosion has a tendency to increase from northwest to southeast [59,60]. Therefore, the phased and spatial characteristics in runoff and sediment load with a decreasing trend at the four stations are important to note. The years from 1960 to 1968 were all high-flow years with few exceptions, and the average annual runoff were 28.28 billion m$^3$, 35.27 billion m$^3$, 47.12 billion m$^3$ and 52.84 billion m$^3$, respectively. The years from 1969 to 1974 were basically consecutive normal flow years. Differently, dry years were obvious from 1990 to 2015 (except 2012) due to the reduction of precipitation and significant utilization of water resources [24,49]. The corresponding annual average runoff rates at the four stations were 16.16 billion m$^3$, 19.05 billion m$^3$, 23.84 billion m$^3$ and 25.54 billion m$^3$. The maximum runoff reached 4.30, 4.07, 4.68 and 6.04 times as much as the minimum values, respectively. The variation of sediment load at the Toudaoguai station was similar to the runoff, while sediment at the other three stations in the four stages of 1960–1980, 1981–1986, 1987–1998 and 1999–2015 showed significant stepwise changes. In the 1980s,

the storage of dams and reservoirs dominated [61]. After 1999, the implementation of a large-scale "returning farmland to forest (grass)" policy became the main reason for the reduction of sediment [24]. The operation of the Longyangxia in 1986 reduced the quantity of runoff and increased the sediment carrying capacity, resulting in an increase in SSC from 1986 to 1999 [59]. The change in degree of runoff and sediment was unevenly distributed in space and the runoff and sediment reductions were not synchronized. Compared with 1960–1968, the annual average runoff of the four stations in 2000−2015 decreased by 41.77%, 47.36%, 50.76% and 51.84%, respectively. The corresponding reduction in sediment load were 78.05%, 86.97%, 82.05% and 91.58%, respectively. The decrement increased along the way. The maximum values were 18.73, 65.08, 44.58 and 159.07 times as much as the minimum value, respectively; thus, the change in annual sediment load was much greater than that in runoff. It is deduced that the impact of human activities especially for soil and water conservation measures on sediment load is more significant than on runoff [62]. It is of great significance for the establishment of regulation system of water and sediment in the MYRB, implement of soil and water conservation measures according to local conditions and design of major water conservancy projects to master the change rules of water and sediment and their relation at the space-time scale. Therefore, there was a significant decrease of correlation between Q and SSC at the Longmen, Tongguan and Huayuankou stations after 2000. From the analyzed data (Figure 7), the accumulated sediment load of Tongguan was the highest in all three stages, resulting in serious silt deposition. Abrupt changes occurred in 1997 at Huayuankou due to regulation of the Xiaolangdi reservoir. Due to the flood retardation and sediment release of Sanmenxia from 1965 to 1973 and the operation of storing clear water and releasing muddy water (with the inputs and outputs balanced) since 1974 [63], the slopes of the cumulative sediment load curves at the Tongguan and Huayuankou stations were basically parallel from 1974 to 1999. Compared with the runoff during 1960–1964, the runoff of Huayuankou from 1965 to 1973 and from 1974 to 1997 decreased by 19.49% and 31.04%, respectively, and the sediment load increased by 74.58% and 8.77%, respectively.

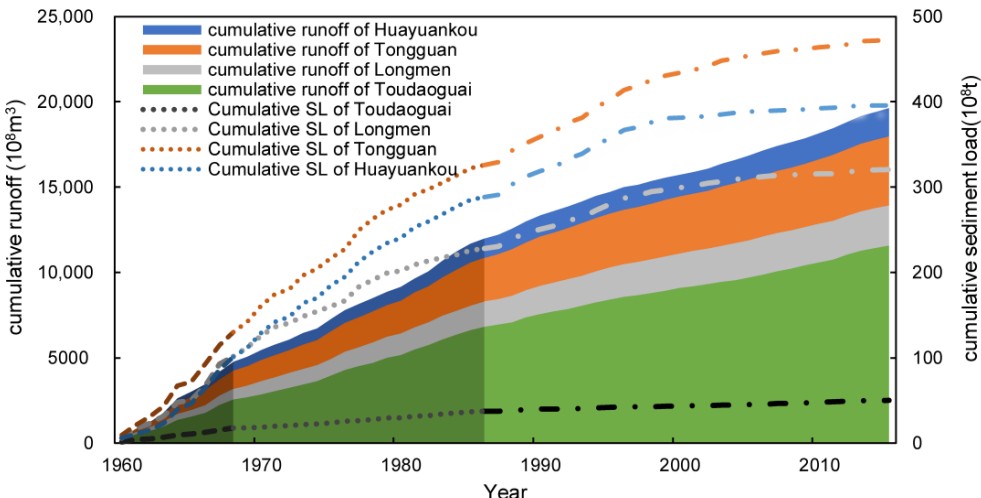

**Figure 7.** Cumulative runoff and sediment load (SL) from 1960 to 2015.

## 5. Conclusions

The current study presented definite equilibrium relationships among SL, SSC and runoff series of the Toudaoguai, Longmen, Tongguan, Huayuankou stations at different temporal scales. These logarithmic variables satisfied linear equilibrium relationships at the Toudaoguai station at multiple time scales, because the variation features of runoff and sediment were similar. At the other stations, the nonlinear cointegration was appropriate at the annual scale. The test of cointegration relationships reveals the quantitative relationship among SL, runoff and SSC and provides a theoretical equation for the calculation and prediction of hydrological variables. In addition, the existence of

the cointegration relationship eliminates the pseudo-regression phenomenon in the subsequent study of the uncertainty relationship between variables. ECM, based on a linear cointegration equation, could improve the prediction accuracy compared with OLS. In general, ECM based on three variables was the best and worthy of recommendation in the forecasting of hydrological variables.

The different hydrological conditions induced by the variations of geomorphologic agent, available sediment, reservoir regulation and other natural and human factors result in uncertain relationships between runoff and sediment. The research on the uncertainty relationship between water and sediment of the Yellow River is of great significance to the construction of the water and sediment control system in the MYRB and the design of major water conservancy projects. The relation between discharge and SSC at the Toudaoguai station could be described by a cubic polynomial function at the within-event scale. Nevertheless, at other three stations, low energy flow with limited fine sediments resulted in a lower SSC with respect to $Q < 1000$ m$^3$/s and emerged rating curves in the form of a power-law function. The SL, runoff and SSC satisfied the linear cointegration relationship at multiple time scales; When $Q > 1000$ m$^3$/s, energy flow with different particle size ratio promotes the formation of hyperconcentrated flow. The SL, runoff and SSC satisfied the linear cointegration relationship at multiple time scales. Meanwhile, the SRCs between the maximum SSC values for a given discharge displayed a linear form once the flow was saturated and stable. Therefore, it is helpful for the calculation of the maximum SSC value for a particular discharge. Meanwhile, effective desilting can be achieved by controlling the volume of discharge (threshold: 1000 m$^3$/s) and the proportion of sediment particle size. Different types of hysteresis loops occurred in different proportions at the four stations due to the different sources of runoff and sediment. At the monthly-seasonal scale, the sediment rating curve derived from a linear regression fitted the data well in the ice-flood season. The rating curve at the Toudaoguai station could also be described by a cubic polynomial function during the rainy season, but the correlations between SSC and Q were very low from Longmen to Huayuankou stations, due to a complex supply of sediment and the formation of hyperconcentrated flow. The hysteresis at the Toudaoguai station exhibited figure-eight patterns, and hysteresis loops with clockwise trends were found for the other stations at a monthly scale. At the annual scale, phased and spatial characteristics in the runoff and sediment load were evident. In addition, the degree of change in runoff and sediment was unevenly distributed in space and the runoff and sediment reduction were not synchronized, because of the impact of human activities on sediment load, which was more significant than on runoff.

**Author Contributions:** Conceptualization, X.W.; Methodology, X.W.; Software, D.L.; Formal Analysis, X.W. and D.L.; Data Curation, X.Q.; Writing—Original Draft Preparation, D.L. and X.Q.; Writing—Review & Editing, X.W. and X.Y.; Supervision, X.W.; Project Administration, P.Z.; Funding Acquisition, X.Y. All authors have read and agreed to the published version of the manuscript.

**Funding:** This research was funded by the National Key Research and Development Program of China (grant number 2018YFC1508403), the National Natural Science Foundation of China (grant number 51579173), the National Key Research and Development Program of China (grant number 2016YFC0401701) and the Science Fund for Creative Research Groups of the National Natural Science Foundation of China (grant number 51621092).

**Acknowledgments:** We wish to express our deep gratitude to the Yellow River Conservancy Commission for permission to access to the hydrological data. Meanwhile, we wish to acknowledge the assistance of all editors and reviewers and thank Sara J. Mason, MSc, from Liwen Bianji, (www.liwenbianji.cn/ac), for editing the English text of a draft of this manuscript.

**Conflicts of Interest:** The authors declare no conflict of interest.

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
