# Peer review of "Analysis of Runoff-Sediment Cointegration and Uncertainty Relations at Different Temporal Scales in the Middle Reaches of the Yellow River, China"

_water, doi:10.3390/w12092589_

Round 1
Reviewer 1 Report
The manuscript submitted for review presents statistical modeling of equilibrium relationship among sediment load, suspended sediment concentration and runoff series of Toudaoguai, Longmen, Tongguan, Huayuankou stations in the middle Yellow River basin (China) at different temporal scales.
The authors used data total 285 floods 1960 to 2015, mostly collected from the Yellow River Water Conservancy Commission.
In Chapter 1 and 2, Introduction and Study area and Materials, the authors explain the topic by citing literature and present the analyzed area and its conditions. 38 references are cited here. Reviewer has no comments.
In Chapter 3, the authors present the Methods. Cointegration is presented very shortly. What is Johansen test? This is linear cointegration. For the nonlinear cointegration was used Wavelet-neural network and typical function. For runoff-sediment relationship, a linear function was used to relate the intensity of sediment to its concentration. In Equation 3, it should probably be SSC, not C.
Chapter 4 presents a results and extensive discussion of the conducted research. Too few references to literature!!! Such a discussion must be supported by literature. It needs to be improved!!! Enlarge figures 2-7.
In chapter 5, the authors present summary main goals of conclusion. Properly. The reviewer has no comments.
Reviewer 2 Report
See attached review.

Round 2
Reviewer 1 Report
All comments were taken into account
Reviewer 2 Report
Several minor grammar/editorial changes are recommended. See review provided.
